# Centering the Margins: Outlier-Based Identification of Harmed Populations in Toxicity Detection

**Vyoma Raman**[*]
UC Berkeley
vyoma.raman@berkeley.edu

**Eve Fleisig**[*]
UC Berkeley
efleisig@berkeley.edu

**Dan Klein**
UC Berkeley
klein@berkeley.edu

## Abstract

The impact of AI models on marginalized communities has traditionally been measured by identifying performance differences between specified demographic subgroups. Though this approach aims to center vulnerable groups, it risks obscuring patterns of harm faced by intersectional subgroups or shared across multiple groups. To address this, we draw on theories of marginalization from disability studies and related disciplines, which state that people farther from the norm face greater adversity, to consider the "margins" in the domain of toxicity detection. We operationalize the "margins" of a dataset by employing outlier detection to identify text about people with demographic attributes distant from the "norm". We find that model performance is consistently worse for demographic outliers, with mean squared error (MSE) between outliers and non-outliers up to 70.4% worse across toxicity types. It is also worse for text outliers, with a MSE up to 68.4% higher for outliers than non-outliers. We also find text and demographic outliers to be particularly susceptible to errors in the classification of severe toxicity and identity attacks. Compared to analysis of disparities using traditional demographic breakdowns, we find that our outlier analysis frequently surfaces greater harms faced by a larger, more intersectional group, which suggests that outlier analysis is particularly beneficial for identifying harms against those groups.

## 1 Introduction

Society often erects barriers that hinder marginalized individuals from receiving essential social and infrastructural access. Disability studies offers valuable insights into their experiences, illuminating the oppressive and restrictive nature of the construction of "normalcy" within society. This rich body of literature highlights how prevailing

---

[*]Eve and Vyoma co-created the theoretical framework for this paper, and Vyoma implemented it.

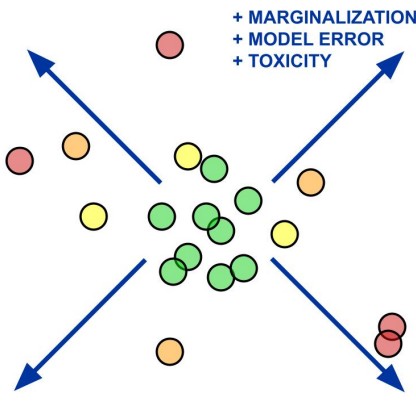

Figure 1: Outliers on the basis of demographic identity face harms resulting from high model error compared to non-outliers. Further analysis of the attributes of demographic outliers can reveal the specific subgroups experiencing harm.

social structures marginalize certain groups and further perpetuate their exclusion from mainstream societal participation (Davis, 1995, 2014). Such arguments are also made in gender studies (Butler, 1990) and related disciplines (Goffman, 1963). Extending the scope of these critiques to the realm of artificial intelligence, we recognize that AI models also encode prevailing notions of normalcy. When models optimize for an aggregate metric, they prioritize patterns and distributions for data points with more common characteristics; that is, the "norm" of the dataset. As such, individuals who fall outside the normative boundaries of the training data are more likely to be subject to model error and consequent harm. The lack of representation and tailored accommodation for these marginalized groups within the training data contributes to biased and exclusionary AI models.

In the evolving landscape of powerful tools that use machine learning, it is crucial to critically evaluate their application to avoid reinforcing systemic biases and instead promote equitable outcomes. Algorithmic auditing plays a vital role in assessing the

real-world impact of AI, especially in identifying and scrutinizing potential harms to specific demographic subgroups and their intersections (Mehrabi et al., 2021; Raji et al., 2020). However, the current subgroup-based analysis used in algorithmic fairness evaluations is fraught with challenges. Two notable concerns emerge: First, identifying the relevant subgroups of concern is not always straightforward (Kallus et al., 2022), as there may be hidden or overlooked patterns in how populations are affected by algorithmic harms. For example, dividing into individual racial subgroups may conceal shared harms experienced by multiracial individuals or people with different races. Second, when considering intersectional subgroups across multiple demographic categories, including race, gender, sexual orientation, disability, religion, and age, the sheer number of potential subgroups becomes overwhelming while the size of each subgroup decreases; thus, particularly severe or unique harms faced by many smaller, intersectional subgroups may be overlooked (Kong, 2022). These limitations make it challenging to conduct thorough and accurate audits.

Inspired by the argument in disability studies that those who fall outside the norm experience greater adversity, we propose using outlier detection to statistically quantify who is assumed to be "normal" and analyze algorithmic harms with respect to this boundary in the domain of algorithmic content moderation. We find that disaggregating harm by outlier group membership reveals high model error disparities compared to other schemes for breaking down the data. Additionally, we identify discrepancies in toxicity detection model performance between outliers and non-outliers and analyze these groups to determine the demographic makeup of those experiencing the most harm. This approach allows for the identification of subgroups and intersections thereof that are particularly vulnerable to harm in the model's behavior.

This paper presents three primary contributions to advance the understanding of algorithmic harm towards marginalized groups in toxicity detection:

1. We uniquely apply existing outlier detection techniques to propose and implement a new approach for identifying marginalized groups at risk of AI-induced harm, translating the "other" examined extensively in social science literature into statistical terms.

2. We evaluate the degree of harm revealed by our outlier-based analysis compared to traditional demographic group-based analyses, and find that demographic and text outliers have a consistently high information yield on model performance disparities.

3. We examine model performance disparities between outliers and their complements to highlight the detection of severe toxicity and identity attacks as areas of concern and identify demographic subgroups that are particularly susceptible to harm from disparities in toxicity predictions.

Our work underscores the importance of incorporating critical theory into auditing practices in machine learning models to ensure they do not exacerbate societal biases or inadvertently harm marginalized communities. The methodologies and tools we present serve as resources for those seeking to create more equitable and inclusive AI systems.

## 2 Prior Work

We ground our work on prior research on algorithmic content moderation and its potential harms, the measurement and evaluation of these harms, and the relationship between outlier detection and the social construction of the "other."

### 2.1 Harms in Content Moderation

Allocative harms in algorithmic toxicity detection occur when content moderation decisions disproportionately amplify or suppress content by or about specific groups. These can also veer into representational harms, which involve the systematic silencing or misrepresentation of marginalized groups (Crawford, 2017; Butler, 1997). For example, toxicity detection algorithms have disproportionately flagged content from minority communities (Hutchinson et al., 2020) and failed to adequately address hate speech targeted at these groups (Binns et al., 2017).

In toxicity detection settings, classifiers are prone to label dialects like African American English as abusive more often, creating a discriminatory effect in content moderation (Halevy et al., 2021; Davidson et al., 2019; Sap et al., 2019). A contrasting challenge lies in correctly identifying implicit hate speech, which frequently manifests through coded or indirect language (ElSherief et al., 2021; Waseem et al., 2017). In response to these

issues, researchers have taken approaches that include proposing conceptual formalisms to capture the subtle ways social biases and stereotypes are projected in language (Sap et al., 2020) and designing benchmark datasets to test the performance of algorithmic systems in fair and explainable toxicity detection (Mathew et al., 2021).

## 2.2 Measuring Algorithmic Harms

The measurement of harms caused by content moderation AI has been approached from ethical, legal, and computational perspectives (Mittelstadt et al., 2016; Barocas and Selbst, 2016; Sandvig et al., 2014). On the computational side, fairness metrics that seek to capture different aspects of algorithmic performance often employ demographic disaggregation to highlight potential disparities and biases that may not be evident in aggregate performance measures (Dwork et al., 2012; Hardt et al., 2016). While these metrics have been critiqued for overlooking underlying differences between groups or confounding intersectional harms (Corbett-Davies and Goel, 2018; Kearns et al., 2018), the overall emphasis on protected groups has remained popular. A focus on disaggregated analysis has enabled a more nuanced understanding of algorithmic harms and informed the development of fairer and more inclusive AI systems (Barocas et al., 2021; Buolamwini and Gebru, 2018).

## 2.3 Qualitative and Quantitative Representations of the "Other"

Disability studies provides a valuable perspective on understanding the social construction of the "other." Some scholars have argued that disability is a social phenomenon as well as a biological category (Shakespeare, 2006). They contend that all bodies and minds are part of a spectrum of natural human diversity and that the distinction between disabled and nondisabled arbitrarily divides "normal" and "abnormal" embodiment and behavior in a harmful way (Davis, 2014). Social categories thus distinguish an in-group ("us") from an outgroup ("them"), favoring the dominant group and marginalizing the "other" (Said, 1988). These principles apply to groups marginalized along axes like race, ethnicity, gender, and sexual orientation, in addition to ability (Goffman, 1963; Butler, 1990).

The concept of "normalcy" emerged alongside these social constructs, heavily influenced by the rise of statistical methodologies that reified the distinction between normal and abnormal (Davis,

1995). This was seen in early applications like IQ tests (Fass, 1980) and phrenology (Twine, 2002), which attempted to justify racism and ableism using pseudoscience. Scholars have argued that all systems discard matter and and reinforce certain structures (Lepawsky, 2019; Thylstrup and Talat, 2020). Modern statistics has further advanced methods of "othering" through the notion of outliers, observations in a dataset that deviate significantly from the norm. Outlier detection involves statistical methods to identify deviations from the norm and understand the situatedness of knowledge, which can be used to approximate the dichotomy between the "norm" and the "other" for research purposes (Haraway, 1988).

By applying outlier detection to data, it is possible to identify minoritized points that represent marginalized people. Groups that are further from the social "norm" face greater social harms, and those with lower representation in datasets may similarly face greater allocative harms since a model will perform worse on data points it had less exposure to during training.

## 3   Methods

Current quantitative methods for determining the impact of content moderation systems on vulnerable populations rely on identifying demographic subgroups represented in the dataset and examining the model behavior toward these groups. However, this approach faces two primary barriers. First, dividing data by demographic group membership allows focus on model harm toward that group, but it can obscure insight into cross-group harm patterns. Second, intersectional harms can be particularly acute and are crucial to measure. This has typically involved a disaggregated analysis of individuals along different demographic categories like race and gender (Buolamwini and Gebru, 2018). However, as more and more demographic categories are considered, the number of subgroups increases exponentially and their size decreases exponentially, making it challenging to identify and address specific problem areas. To mitigate the risk of missing significant harms, we propose a model harm identification framework to address these limitations and determine which groups and subgroups experience poorer model performance.

### 3.1 Model and Data

In this study, we examined the impact of algorithmic content moderation on various demographic groups by conducting our analyses on three free and publicly available toxicity and hate speech classifiers: the Perspective API, a toxicity detection tool developed by Jigsaw (Zhang et al., 2018), an ELECTRA model fine-tuned at the University of Tehran on HateXplain (Modarressi et al., 2022; Mathew et al., 2021), and a RoBERTa model fine-tuned at Meta on dynamically generated datasets (Vidgen et al., 2021). Both of the latter are released on HuggingFace. The Perspective models predict several toxicity-related attributes, including toxicity, severe toxicity, identity attack, insult, obscenity, and threat, each associated with specific definitions provided by Jigsaw (Appendix A). The other two models simply predict hate speech.

We selected the Jigsaw Unintended Bias dataset, available publicly on Kaggle and HuggingFace, due to its detailed demographic annotations (Adams et al., 2019). Its toxicity annotations are also calibrated with with the Perspective models, but not the other ones. The dataset contains comments that were collected by Civil Comments, a plugin for improving online discourse on independent news sites, and released when Civil Comments shut down in 2017. Jigsaw applied a framework from the Online Hate Index Research Project to annotate these comments based on the Perspective API's attributes and 24 demographic groups within race, gender, sexual orientation, religion, and disability (onl, 2018). This was done by collecting 4-10 binary annotations for the presence of each topic for each comment in the dataset and averaging them to come up with a decimal score.

The full dataset with demographics annotations consists of 445,294 comments. Due to computational resource constraints, we applied stratified sampling to select 10% of the rows with each demographic group and removed duplicate rows. We obtained model predictions from each toxicity category from Perspective and for the general toxicity category from the other two models. Each row in our dataset contains information about one comment, including the comment text and binary and decimal representations of the toxicity attribute, demographic groups mentioned in the text, and annotator disagreement. We converted decimal representations to binary ones using a threshold of 0.5. We computed the annotator disagreement binary values by identifying whether the averaged value was 0 or 1 (indicating that all the annotators had the same opinion) or whether it was somewhere in between (indicating different labels). We computed the annotator disagreement decimal values by taking the variance of the score of the averaged score from annotators. The final dataset comprised 20,589 rows and 180 columns.

### 3.2 Outlier Detection and Analysis

We used the Local Outlier Factor (LOF) method for outlier detection. LOF measures the deviation in local density of a point from its neighbors. This involves computing the inverse of the average distance from a point to its nearest neighbors and using a threshold to determine whether the point is an outlier (Breunig et al., 2000). To have standardized volumes of different outlier types for comparison purposes, we set the contamination parameter to 0.05.[1] This automatically selects the threshold such that 5% of points are outliers. LOF's calculation of data point density serves as a quantitative representation of the "norm," with the selected outliers becoming the "other."

We implemented LOF on multiple vector sets to examine different types of outliers: text-based, disagreement-based, and demographic-based. We use a contamination value of 0.05, selecting 5% of the points as outliers. We specified a consistent contamination value instead of an explicit threshold when running LOF to have consistent proportions of outliers (5%) across the three outlier types. The resulting thresholds were -0.981 for demographic outliers, -0.989 for text outliers, and -0.982 for disagreement outliers.

1. Text outliers are determined using Doc2Vec embeddings of the text of each comment. In this context, outliers are comments with unusual words, phrases, topics, syntax, or other textual patterns.

2. Disagreement outliers are determined from a vector of annotator disagreement values, computed by taking the variance of the score of the averaged score from annotators. These indicate comments for which annotators clashed in unusual ways.

---

[1]All data manipulation and analysis were performed using the Pandas, Scikit-Learn, and Gensim libraries (McKinney, 2010; Pedregosa et al., 2011; Rehurek and Sojka, 2011).

3. Demographic outliers are computed using vectors of demographic labels annotated on the dataset, and they represent comments that discuss an unusual demographic group or set of groups.

We expected model performance for all these types of outliers to be poorer than for non-outliers: text-based outliers may contain unusual linguistic harm patterns that are not recognized by the model or reclaim offensive language in positive ways; disagreement-based outliers may simply be more ambiguous in their toxicity than non-outliers; and demographics-based outlier comments may include mentions of groups or include demeaning content that the model has less exposure to.

For each outlier type, we examined significant differences in average toxicity between outliers and non-outliers to identify any disparities in the experiences of these groups. To determine if this pattern is pervasive throughout the dataset, we repeated the analysis on each demographic subgroup. This approach allowed us to gain a deeper understanding of the impact of toxicity both generally and on various demographic groups within the context of outliers and non-outliers.

In Section 4.4, we also analyzed associations between general and intersectional demographic characteristics. We inspected the proportion of members of demographic subgroups considered outliers and non-outliers to determine the distribution of outliers within each demographic subgroup. We also examined the average number of demographic identities in outlier and non-outlier points. These analyses helped us to understand trends in the association of individual identities or intersections of demographic characteristics with outlier classification, which can help to further assess the potential impact of AI models on these populations.

## 3.3 Model Performance Disparity Scoring

Comparing model performance between outliers and non-outliers allowed us to assess the impact of the model on various groups, considering both overestimation and underestimation of toxicity. We stratified the results by demographic subgroups to determine the pervasiveness of different issues.

When determining what harms particular subgroups face, a critical step is breaking the dataset down into subgroups to identify the ones facing increased harms. To experimentally test our hypothesis that outlier-focused analysis spotlights the subpopulation of marginalized individuals facing greater model harms, we compare model performance across in-groups and out-groups for several subpopulations in the dataset.

For a binary demographic attribute $i$ and a full set of data points $D$, the set $g_i$ is a subset of $D$ whose elements have attribute $i$:

$$g_i = \{x \in D : x_i == True\}$$

When splitting the dataset by group membership, we measure the disparity in model performance toward a group $g_i$ as the relative difference in the model's mean squared error between $g_i$ and its complement for each toxicity type, weighted by how often the group experiences that form of toxicity (abbreviated $\text{WMSE}_{g_i}$ for convenience):

$$\text{WMSE}_{g_i} = \sum_{t \in T} freq(g_i, t) \frac{\text{MSE}(g_i, t) - \text{MSE}(\neg g_i, t)}{\text{MSE}(\neg g_i, t)}$$

A positive $\text{WMSE}_{g_i}$ indicates that model performance is generally worse for members of the group than for non-members, while a negative score indicates that it is better. Since we are investigating model harm toward a particular group, we focus on positive $\text{WMSE}_{g_i}$.

We are interested in understanding whether our outlier analysis reveals patterns of harm not visible in analyses based solely on membership in a marginalized demographic, so we compare the outlier analysis to several alternatives:

1. Marginalized group membership: Whether the combined set of all marginalized groups along a demographic axis faces higher harms. We consider people of color (Black, Latine, Asian, or "other race"), gender minorities (women and "other gender identity"), sexual minorities (gay, lesbian, bisexual, or "other sexual orientation"), U.S. religious minorities (atheist, Buddhist, Hindu, Jewish, Muslim, and "other religion"), and disabled people (intellectual, physical, psychiatric, and "other disability").[2]

2. Binary demographic group membership: Whether groups face higher harms on the basis of single binary demographic attributes (e.g., "female" or "Black").

---

[2] We note as a limitation that these categories are certainly not exhaustive of all identities that are marginalized along these axes, due to the restricted options for identity categories in the Jigsaw dataset.

3. Intersectional demographic group membership (2 groups): Whether groups face higher harms on the basis of two binary demographic attributes (e.g., "Black women" or "bisexual men").

By comparing the $\text{WMSE}_{g_i}$s of various group breakdowns, we can assess the value of considering outlier status in the analysis of model performance. Compared to these analyses, we expected that outliers would be among the largest groups facing the most severe harms.

## 4 Results

We used outlier detection to identify outliers on the basis of text, demographics, and annotator disagreement. Approximately 1,000 samples were labeled as outliers for each outlier type since we set the contamination parameter in the Local Outlier Factor (LOF) algorithm to 0.05. We set the n_neighbors parameter to 4,000 based on the size of the dataset. We compared outliers and non-outliers by employing statistical testing to determine the significance of metric differences. We used the $\chi^2$ test of homogeneity to compare group average scores, complemented with a Bonferroni correction.

### 4.1 Relative Difference in Mean Squared Error

Before analyzing model performance between outlier groups and their complements, we sought to understand how much disparity in model performance is revealed by different types of group breakdowns of the dataset. To do this, we examined the relative difference in mean squared error, as described in Section 3.3, across different types of disaggregation.

| Percentile Group | Demographic Outliers | Text Outliers | Disagreement Outliers |
|---|---|---|---|
| Marginalized | **100**% | 44.4% | 33.3% |
| Binary | **92.6**% | 81.5% | 63% |
| Intersectional | **88.5**% | 83.2% | 78.9% |

Table 1: Demographic outliers have a consistently high $\text{WMSE}_{g_i}$ in the Perspective models.

Tables 1, 2, and 3 describe the $\text{WMSE}_{g_i}$s for each group across the three models for three schemas for demographic breakdowns: marginalized group membership, binary group membership, and intersectional group membership. Notably, demographic outliers have among the high-

| Percentile Group | Demographic Outliers | Text Outliers | Disagreement Outliers |
|---|---|---|---|
| Marginalized | 33.3% | **77.8**% | 44.4% |
| Binary | 51.9% | **92.6**% | 66.7% |
| Intersectional | 21.1% | **95.0**% | 83.9% |

Table 2: Text outliers have a consistently high $\text{WMSE}_{g_i}$ in the RoBERTa model.

| Percentile Group | Demographic Outliers | Text Outliers | Disagreement Outliers |
|---|---|---|---|
| Marginalized | 44.4% | **88.9**% | 33.3% |
| Binary | 74.1% | **81.5**% | 59.3% |
| Intersectional | 83.2% | **85.7**% | 79.6% |

Table 3: Text outliers have a consistently high $\text{WMSE}_{g_i}$ in the ELECTRA model.

est percentile of $\text{WMSE}_{g_i}$ for all three breakdowns for the Perspective models. Figure 2 depicts the $\text{WMSE}_{g_i}$ for the Perspective models' marginalized group membership breakdown and Figure 3 does so for the binary demographics. On the other hand, text outliers have among the highest percentile of $\text{WMSE}_{g_i}$ across the breakdowns for the ELECTRA and RoBERTa models. These collectively illustrate that outlier-based disaggregations have the highest information yield, with different types of outliers being more influential in discovering disparities in $\text{WMSE}_{g_i}$ for different models. These results demonstrate the importance of varied and strategic data breakdowns in uncovering potential model harms.

### 4.2 Toxicity Analysis

Our dataset reveals a pervasive trend where general toxicity (12.2% frequency), identity attack (4.89%), and insult (5.62%) emerge as the most common forms of toxic speech. This is consistent across demographic subgroups.

We uncovered noticeable differences in how outliers and non-outliers experience various forms of toxicity. Figure 4 illustrates how identity attack (86%, 58.4%), severe toxicity (64.1% and 40.7%), and general toxicity (40.2% and 24.8%) are significantly more severe for demographic and text outliers.

In contrast, we observed that toxicity, identity attack, and insult disparities are negative between disagreement outliers and non-outliers, indicating less toxicity in disagreement outliers. In examining potential reasons for this trend, we found that disagreement outliers have higher agreement than

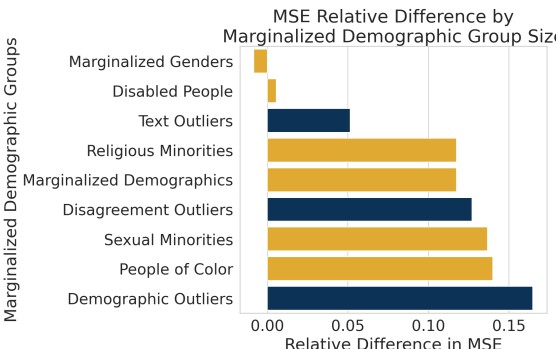

Figure 2: For the Perspective models, demographic outliers, when compared with nine different subgroups, demonstrated the highest $\text{WMSE}_{g_i}$, suggesting that it is most effective at uncovering which groups face the greatest disparities.

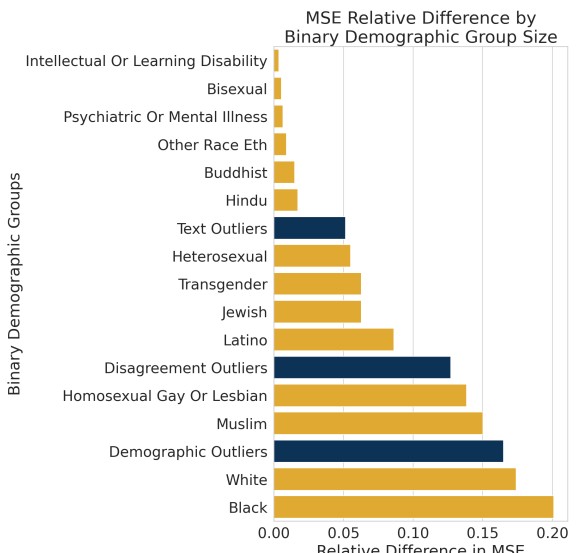

Figure 3: A comparative analysis reveals that demographic outliers have among the highest $\text{WMSE}_{g_i}$ of 27 different subgroups (17 with positive values pictured) for the Perspective models.

non-outliers (32.8-49% more) for these toxicity types compared to other types (17.9-20.7% more). This suggests that disagreement outliers contain comments that are widely viewed as harmless.

Moreover, when verifying our results, we found differences in scores of points in and out of each outlier group. To understand the pervasiveness of these differences, we counted the number of demographic groups where differences in toxicity scores between outliers and non-outlier points within those groups were significant (Table 8). Consistent with the trend across demographics, the most subgroups experience significant disparities between outliers and non-outliers for identity at-

| Toxicity Type | Demographic Outliers | Text Outliers | Disagreement Outliers |
|---|---|---|---|
| Identity Attack | 8 | 6 | 11 |
| Toxicity | 7 | 7 | 15 |
| Insult | 7 | 5 | 13 |
| Severe Toxicity | 4 | 3 | 1 |
| Obscenity | 5 | 2 | 1 |
| Threat | 1 | 1 | 1 |

Table 4: Number of demographic groups with a statistically significant difference in scores for a particular toxicity type for each definition of outliers.

| Toxicity Type | Overall MSE | Outlier MSE | Non-Outlier MSE | MSE % Increase on Outliers |
|---|---|---|---|---|
| Identity Attack | 0.030 | 0.049 | 0.029 | 70.4% |
| Severe Toxicity | 0.002 | 0.003 | 0.002 | 59.1% |
| Threat | 0.006 | 0.008 | 0.005 | 41.0% |
| Toxicity (Perspective) | 0.032 | 0.043 | 0.032 | 35.6% |
| Obscenity | 0.009 | 0.012 | 0.009 | 33.7% |
| Insult | 0.022 | 0.028 | 0.022 | 29.5% |
| Toxicity (ELECTRA) | 0.06 | 0.067 | 0.059 | 14.0% |
| Toxicity (RoBERTa) | 0.127 | 0.125 | 0.127 | -2.06% |

Table 5: Model performance overall and divided by demographic outlier status across all types of toxicity.

tack, general toxicity, severe toxicity, and insulting harms. Identity attacks are experienced most intensely (86% worse) and pervasively (significant differences in 32% of subgroups) by demographic outliers as well as by text outliers. This suggests that demographic outliers may be particularly insightful for understanding the harms toward different subgroups of people in the dataset.

### 4.3 Model Performance Analysis

For the remaining results, we focus on demographic outliers for the Perspective models and text outliers for the ELECTRA and RoBERTa models, since they have consistently high $\text{WMSE}_{g_i}$ percentiles. These scores for different outlier types underscore the high degree to which the identification of outliers can expose model harm. Additionally, their heightened degree of identity attacks makes them a particularly valuable group to study with respect to their unusual identity characteristics.

In our analysis, we examined the model perfor-

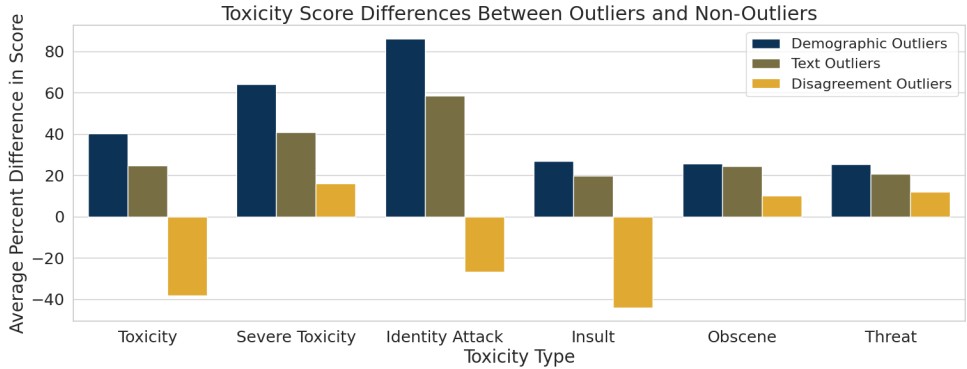

Figure 4: Average differences in ground truth toxicity between three different types of outliers and their complements. Identity attack and severe toxicity have the greatest significant differences. Demographic outliers consistently face the highest toxicity; disagreement outliers show the opposite trend.

| Toxicity Type | Overall MSE | Outlier MSE | Non-Outlier MSE | MSE % Increase for Outliers |
|---|---|---|---|---|
| Severe Toxicity | 0.002 | 0.003 | 0.002 | 68.4% |
| Identity Attack | 0.030 | 0.040 | 0.029 | 37.2% |
| Toxicity (RoBERTa) | 0.127 | 0.169 | 0.125 | 35.2% |
| Toxicity (ELECTRA) | 0.06 | 0.072 | 0.058 | 22.9% |
| Obscenity | 0.009 | 0.010 | 0.008 | 15.1% |
| Threat | 0.006 | 0.006 | 0.005 | 14.3% |
| Insult | 0.022 | 0.024 | 0.022 | 8.34% |
| Toxicity (Perspective) | 0.032 | 0.034 | 0.032 | 5.03% |

Table 6: Model performance overall and divided by text outlier status across all types of toxicity.

mance across different toxicity types for demographic outliers (Table 5) and text outliers (Table 6). The MSE for different toxicity detection scores ranges from 0.002 to 0.032. The steep error for multiple toxicity types shown here is concerning because, depending on the thresholds used for content filtration, it could lead to either the preservation of toxic content or the erasure of benign discourse. With respect to outlier status, we examined the percent difference in MSE between outliers and non-outliers. This ranges from -2.06% to 70.4% for demographic outliers and 5.03% to 68.4% for text outliers. As such, the differences in MSE between outliers and non-outliers for all types of toxicity were positive and significant.

Since the percent differences in MSE are generally positive, the models have more error in the outlier group than in the non-outlier groups, splitting by text outliers and demographic outliers. Notably, disaggregating by both of these outlier types identi-

fies a significant disparity in MSE for the models predicting severe toxicity and identity attacks. This suggests that the models are not performing as well for these subgroups of individuals and implicates them in harming outlier demographic subgroups. This in itself is unsurprising; model performance is likely worse on outliers because, by definition, the model has learned from less data similar to the outliers. However, by identifying which groups are affected by this data constraint, we can determine how to focus efforts to improve models.

### 4.4 Demographic Analysis

Our analysis found that demographic outliers exhibited significant differences in the proportion of each demographic subgroup that is classified as an outlier (Figure 5). More than half of the points of a quarter (6/24) of demographic groups were outliers, including Hinduism, bisexuality, and physical disability. In contrast, a quarter of the groups had any outliers. We also found a significantly higher average number of identities mentioned for demographic outliers compared to non-outliers: 3.7 compared to 1.53. This suggests that individuals belonging to multiple intersectional groups may experience compounded harm from the model.

Our results were slightly different for text outliers. Only a quarter of demographic groups were over 5% outliers, with just the Asian group in more outlier points than non-outlier points. A third (8/24) had no text outliers at all. We also found significantly more identities mentioned for text outliers than non-outliers, 1.75 compared to 1.63. While the associations with demographic groups are milder for text outliers than demographic outliers, they still exist, and this is predictable based on the con-

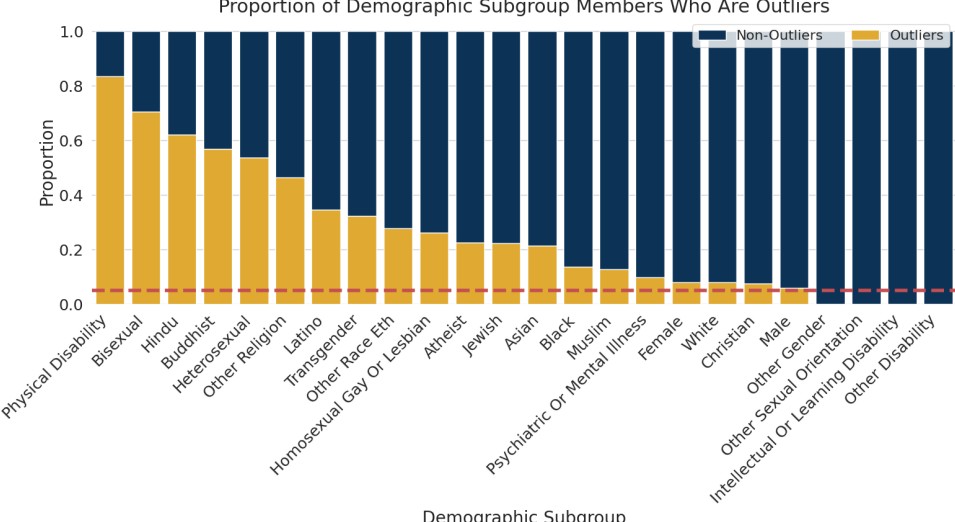

Figure 5: Proportions of each demographic subgroup that are considered outliers or non-outliers. The red line indicates the overall proportion of outliers. Four subgroups are >50% outliers, and four have no outliers.

struction of the outlier group.

By focusing on the demographic makeup of outlier groups with disparities in MSE, we can identify where to apply fairness efforts. Given that mentions of race are overrepresented in text outliers and mentions of minor religions (among other categories) are overrepresented in demographic outliers, harm mitigation for the ELECTRA and RoBERTa models should focus on the former and mitigation for the Perspective models should focus on the latter.

## 5 Conclusion

Our research leverages the insights of disability studies to illuminate unintended harms inflicted by AI systems, particularly those operating in toxicity detection. Societal constructions of "normalcy"—often shaped and reinforced by statistical methodologies—can lead to biases and exclusions in AI models, which in turn amplify societal barriers and inequities. To address this challenge, we have made three contributions. First, we proposed and implemented a method for identifying marginalized groups at risk of AI harm by using outlier detection techniques to identify and examine three different types of outliers. Second, we found that dividing the dataset by outlier status results in consistently high weighted differences in MSE, indicating that our technique successfully exposes serious disparities in model error. Finally, we critically examined model performance disparities across six types of toxicity, finding identity attacks and severe toxicity to be particularly acute

and pervasive for demographic and text outliers. We also identified demographic groups as disproportionately represented among outliers, making them particularly vulnerable to harm from such disparities.

There are two interesting side effects of these results. First, by determining which type of outlier causes the most difference, we can understand how the model responds to differences in syntactic (text features) and semantic (demographic mentions) information. Second, if text-based outliers are insightful for uncovering model harm, model developers can allocate fewer resources to collecting high-quality demographic labels for their dataset, instead focusing on mitigating harms for groups overrepresented in text outliers.

Our research findings and methodologies pave the way for immediate application and future exploration in algorithmic auditing and harm mitigation. A promising direction for future research lies in assessing the use of outlier detection to identify algorithmically harmed groups in scenarios lacking explicit demographic data. This approach could broaden fairness auditing's scope and deepen our understanding of statistical normalcy, social marginalization, and their manifestations in AI systems.

## Limitations

The research in this paper was conducted using the Jigsaw Unintended Bias in Toxicity Classification dataset. This covers English data only, and

information on the geographic, linguistic, and demographic background of the comment writers and annotators was not provided. As such, it is unclear what worldviews the dataset reflects and what types of demographic groups are familiar to the annotators as potential targets of harmful speech.

We selected this dataset for its detailed demographic labels. This helped us probe the demographic identities represented among outliers to determine which subgroups are minoritized and in need of focused harm mitigation efforts. However, such granular labeling may not be present in datasets for other applications. Researchers who do not have additional resources to conduct such labeling and seek to apply these methods in other contexts may thus be limited in the construction and interpretation of demographic outlier groups.

We acknowledge that using a commercial model may hinder reproducibility. We use API version v1alpha1 for this analysis. According to the Perspective API's changelog, the English-language models have not been updated since September 17, 2020, and we completed our analysis in June 2023.

Moreover, the scope of this study was constrained to a single dataset and three models. While we demonstrate the information yield of outliers with respect to model performance disparities in this context, further research is needed to extend our findings to other datasets and tasks. This is further compounded by the sensitivity of outlier detection to the size and schema of a dataset, making our methods contingent on the quality and depth of collection processes.

## Ethics Statement

Our research relies on demographic labels applied to text in the Jigsaw dataset. These labels are an aggregate floating point value representing the proportion of annotators who believed a given identity was represented in a piece of text. The lack of transparency on the backgrounds of the annotators or the data collection process makes it challenging to verify the validity of the labels. We also acknowledge that our method of giving points with a value greater than 0.5 a positive label may obscure the opinions of minority annotators.

Our work proposes using outlier detection as a means of identifying groups potentially harmed by algorithmic bias. This approach has the potential to minimize the need for demographic label inference in future AI fairness evaluations, thus avoiding

potential pitfalls and biases associated with such inferences. However, this method needs further exploration and rigorous testing to confirm its efficacy and examine tradeoffs before being used in high-stakes domains.

Furthermore, outlier detection is frequently used to prune noisy data to improve model performance. This has the inadvertent effect of exacerbating minoritization and othering in the model. Assigning communities to be outside the "norm" can thus cause great harm. While our work seeks to provide a beneficial dual use of outlier detection to support marginalized groups, we acknowledge that this tool can do great harm as well.

## Acknowledgments

Thank you very much to Catherine Chen, Nicholas Tomlin, Eric Wallace, Kevin Yang, and the anonymous reviewers for their thoughtful feedback on this work!

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

| Toxicity Type | Definition |
|---|---|
| Toxicity | A rude, disrespectful, or unreasonable comment that is likely to make people leave a discussion. |
| Severe Toxicity | A very hateful, aggressive, disrespectful comment or otherwise very likely to make a user leave a discussion or give up on sharing their perspective. This attribute is much less sensitive to more mild forms of toxicity, such as comments that include positive uses of curse words. |
| Identity Attack | Negative or hateful comments targeting someone because of their identity. |
| Insult | Insulting, inflammatory, or negative comment towards a person or a group of people. |
| Obscenity | Swear words, curse words, or other obscene or profane language. |
| Threat | Describes an intention to inflict pain, injury, or violence against an individual or group. |

Table 7: Definitions of each toxicity type for the Perspective API.

# A  Jigsaw Unintended Bias in Toxicity Detection Dataset

The full dataset of 445,294 rows contains English data only. To reduce the computational cost of this dataset, we selected a sample (random state=1) of 20,589 rows stratified by demographic labels for our analysis. Additional processing included computing binary versions of demographic floats and ground truth toxicity scores using a 50% threshhold and adding model scores and binary labels from the Perspective API.

# B  Outlier Detection Specifications

Outlier detection was conducted using Scikit-Learn's implementation of Local Outlier Factor. The number of neighbors parameter was set to 4,000, selected based on the size of our dataset. Outliers were computed with 5% contamination, which is standard for outlier detection. This resulted in thresholds of -0.981 for demographic outliers, -0.989 for text outliers, and -0.982 for disagreement outliers.

# C  WMSE$_{g_i}$ by Group Size

We investigated whether outlier groups were better at revealing model harms than demographic groups of a similar size. This was done by computing outliers with varying levels of contamination (0.1,

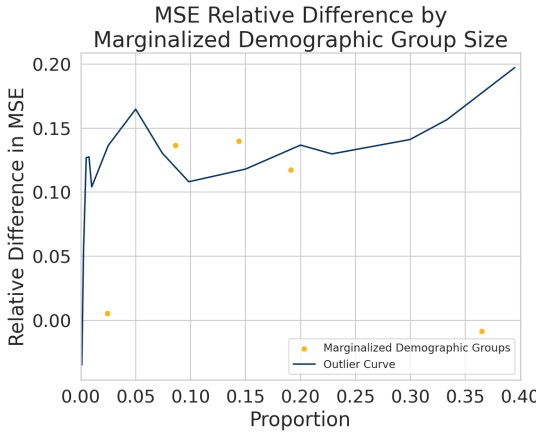

Figure 6: Three of five demographic groups have $\text{WMSE}_{g_i}$s below the curve.

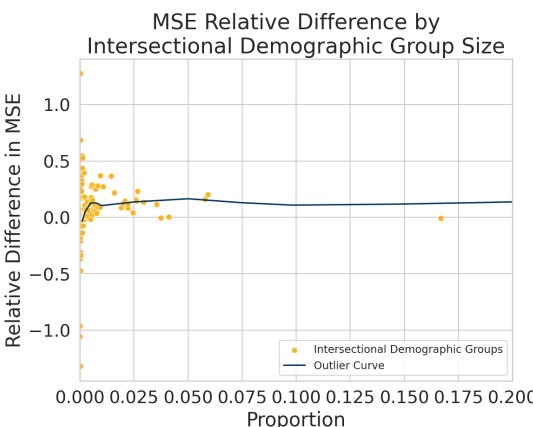

Figure 8: Several of 300 demographic groups have $\text{WMSE}_{g_i}$s below the curve.

## D  Example Text

Content Warning: This table contains text that may be offensive to members of different demographic groups.

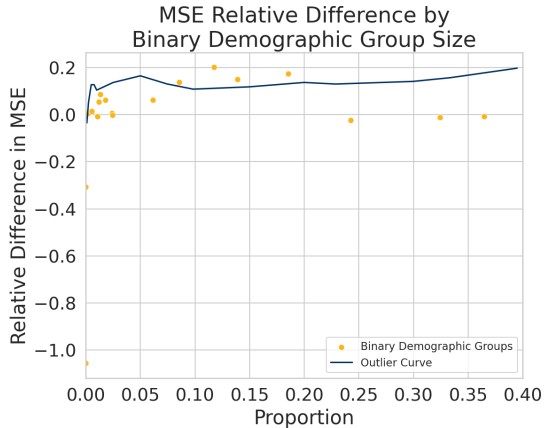

Figure 7: Four of 24 demographic groups have $\text{WMSE}_{g_i}$s below the curve.

0.25, 0.5, 0.75, 1, 2.5, 5, 7.5, 10, 15, 20, 25, 30, 35, 40) up to the size of the largest demographic group in our dataset. We then plotted a curve of outlier $\text{WMSE}_{g_i}$s alongside points for demographic groups to observe how many groups were below the curve.

These figures illustrate how the $\text{WMSE}_{g_i}$s of outlier groups compares to that of other demographic groups. In Figures 6 (marginalized group membership) and 7 (binary group membership), a minority of demographic groups are above the curve. This changes in Figure 8 (intersectional group membership), which illustrates an even divide of demographic groups above and below the outlier curve, an unsurprising finding given the heightened complexity of the problem. It also suggests that the utility of outlier detection for harm measurement varies with the relative size of the outlier population, and that a relative outlier population of 5% (which is typically recommended) may be best.

| Outlier Type | Outlier Comment | Non-Outlier Comment |
|---|---|---|
| Text | There are some interesting numbers coming out of the NY Times post election exit polling. All those who want to claim it was racist white people who elected trump need to look again! Trump only carried 1% more of the White vote than did Romney in 2012 however he received 7% more of the Black vote, 8% more of the Latino vote, and 11% more of the Asian vote. So in reality it was non-White Americans who gave Trump the edge over Crooked Hillary. Uh oh, that doesn't play into the left's narrative though.... | It's a bunch of mostly white guys feigning anger with the NFL and its players because Donald Trump told them to. Trump twisted and contorted the reason the players are protesting, and his followers ate it up. If people want to be forced to stand for the national anthem, North Korea is calling their name. You either support peaceful free speech, or you support the "very fine people" in Charlottesville carrying swastikas and torches. It's that simple. |
| Demographic | So sad that these men were taught they would go to heaven by hurting so many. May need to realize that Islam HAS an internal issue. I don't hear this coming out of Christian, Hindu, Buddhist churches. Maybe with some extremist Jews, but they have also been targeted forever. Hope we learned to be selective on who comes to our country. | "..a moral compass"... lol! I'm sure it was great in the '50s as long as you where white, male, heterosexual and Protestant. |
| Disagreement | Can we have a Pride sort of Fest for straight people? I'm definitely proud of being straight. But is pride and parades only for lgbt? Explain that to the kiddies... | Police are doing their job. This man was in to the store to steal. He is a robbery a criminal. No matter if He was black or white. This man will always steal from every store He goes. Criminal will always a criminal if you are black or white. |

Table 8: Examples of comments relevant to each outlier group and its complement.