# OpenReview forum: "Centering the Margins: Outlier-Based Identification of Harmed Populations in Toxicity Detection"
_EMNLP/2023/Conference — EMNLP 2023 Main_

### Official Review · Reviewer_KWca · 2023-08-05

**Soundness:** 4

**Excitement:**

4: Strong: This paper deepens the understanding of some phenomenon or lowers the barriers to an existing research direction.

**Missing References:**

References are mostly good.

**Paper Topic And Main Contributions:**

The paper attempts to understand how AI models impact marginalized communities. Traditionally, this is handled by examining how models (over- or under-) perform on various subgroups, which ignores intersectional subgroups. This paper avoids this pitfall by taking a quantitative approach (outlier detection) to defining those groups which are most distant from the "norm". Results show that text targeting these groups is especially toxic and model performance is lower.

**Questions For The Authors:**

* What exactly is the data set? You say "comment text" but where the comments posted? Is this Reddit, Twitter, Youtube, etc data (for example)?

"...stratified sampling to select a subset of the data, ensuring a proportional representation of all demographic subgroups..." What exactly does this mean? Subgroups are stratified based on some Census data or something similar?

* Can you be more specific about the "Disagreement-based outliers" vectors? Specifically, what is a "binary vector indicating annotator disagreement"?

* "The final dataset comprised 20,589 rows and 176 columns" Can you provide details are to what this means? For example, what is a row? Is it a single comment or are comments repeated across rows (with each row being an annotation)? Are you using all 176 columns?

* What does the Local Outlier Factor (LOF) method do exactly?

* "Approximately 1,000 samples were labeled as outliers in each method since we set the contamination parameter in the Local Outlier Factor (LOF) algorithm to 0.05." How does 1000 samples labeled as outliers follow from the parameter value?

* Where does Bonferroni correction come in? I don't see any significance levels reported.

**Reasons To Accept:**

Very interesting approach to an important problem. The idea of quantifying distance from the norm is interesting (as opposed to simply splitting subgroups) and it's easy to imagine how this work could be built on. Overall, I enjoyed this paper.

**Reasons To Reject:**

The main weakness is in the lack of description in the data set and methods. I realize the paper uses an open source data set and uses a preexisting method (Local Outlier Factor), but additional details would be nice. I've added clarifying questions below.

**Reproducibility:**

2: Would be hard pressed to reproduce the results. The contribution depends on data that are simply not available outside the author's institution or consortium; not enough details are provided.

**Reviewer Confidence:**

3: Pretty sure, but there's a chance I missed something. Although I have a good feel for this area in general, I did not carefully check the paper's details, e.g., the math, experimental design, or novelty.

**Typos Grammar Style And Presentation Improvements:**

The numbers in the figures are *very* difficult to read.

The color scheme across your figures is consistent but the colors don't represent the same thing. For example, you use the same shade of blue across Figures 2 and 3 (for your Outliers) but this same blue is used for Demographic Outliers (alone) in Figure 4.

---

> ### Author Rebuttal · Authors · 2023-08-29
>
> Thank you very much for your helpful feedback. We will update our description of methods in the camera-ready version of the paper based on your comments. We also answer your questions further below:
>
> **Dataset Description**
>
> The comments analyzed in our dataset were collected by Civil Comments, a plugin for improving online discourse on independent news sites, and released when it shut down in 2017. Jigsaw applied a framework from the Online Hate Index Research Project to annotate these comments based on the Perspective API’s attributes and 24 demographic groups within race, gender, sexual orientation, religion, and disability. This was done by collecting 4-10 binary annotations for the presence of each topic for each comment in the dataset and averaging them to come up with a decimal score.
>
> **Stratified Sampling**
>
> The full set of data with demographics annotations consists of over 400,000 rows. Due to the Perspective API’s rate limits, we chose to select a sample of around 20,000 rows. To ensure that all demographic groups were captured in this sample, we selected 5% of the rows with each demographic group and removed duplicate rows.
>
> **Disagreement Outliers**
>
> A disagreement outlier is an outlier determined using a vector of annotator disagreement values, computed as described below.
>
> **Data Granularity & Usage**
>
> Each row in our dataset describes one comment. Each column contains information about the comment, including the comment text and binary and decimal representations of the toxicity attribute, demographic groups mentioned in the text, and annotator disagreement. We converted decimal representations to binary ones using a threshold of 0.5. We computed the annotator disagreement binary values by identifying whether the averaged value was 0/1 (indicating that all the annotators had the same opinion) or whether it was somewhere in between (indicating different labels). We computed the annotator disagreement decimal values by taking the variance of the score of the averaged score from annotators. While our paper does not reflect an analysis of every single column, we utilized them all to create outlier vectors, disaggregate analysis by demographic groups, and conduct exploratory analyses.
>
> **Local Outlier Factor**
>
> Local Outlier Factor measures the deviation in local density of a point from its neighbors. This involves computing the inverse of the average distance from a point to its nearest neighbors and using a threshold to determine whether the point is an outlier. To have standardized volumes of different outlier types for comparison purposes, we set the contamination parameter to 0.05. This automatically selects the threshold such that 5% of points are outliers. We have updated our manuscript to list the resulting threshold values selected based on this contamination.
>
> **Statistical Testing**
>
> When probing ground truth toxicity labels, we found differences in scores of points in and out of each outlier group. To understand the pervasiveness of these differences, we counted the number of demographic groups where differences in toxicity scores between outliers and non-outlier points within those groups were significant (Table 2). We applied a Bonferroni correction to adjust significance levels by the number of demographic groups.
>
> **Figures**
>
> We will improve the color and text sizing of our figures for clarity.

---

### Official Review · Reviewer_xD9v · 2023-08-05

**Soundness:** 2

**Excitement:**

3: Ambivalent: It has merits (e.g., it reports state-of-the-art results, the idea is nice), but there are key weaknesses (e.g., it describes incremental work), and it can significantly benefit from another round of revision. However, I won't object to accepting it if my co-reviewers champion it.

**Paper Topic And Main Contributions:**

The paper utilizes a standard dataset and performs outlier analysis using the Perspective API as a base model.  WMSE is used as the metric to compare differences among groups and subgroups. The paper seeks to address the pressing and critical issue of intersectionality in NLP-bssed content moderation.

**Questions For The Authors:**

Was the choice of using the Jigsaw dataset along with a model developed by Jigsaw intentional? If so, would love to gauge the thought process and reasoning behind it.

Why limit yourself to the one model and one dataset? Was this a limitation by data, expertise, time?

Did the authors consider the possibility that data limitations (shortages) could potentially lead to data being labeled as an outliers AND having worse performance?



**Reasons To Accept:**

The paper is well written and on a topic which merits more attention by the NLP reseaech community. In particular the paper contains background research and theory which often is overshadowed by  benchmarks in conventional work.

**Reasons To Reject:**

While written well and on theme, the work to my knowledge does not contain substantial novelty. The model used in a commercially available model (Persepctive API). While the relevance of such  a widely used model can be argued, to make general claims about group differences it would be best if the  claims were substiated across multiple models and/or datasets rather than just the one. Finally, using a commercial model also limits reproducibility of the work (the API is an inaccessible black box which are constantly being updated by Jigsaw so results may change over time furthermore the service is likely paid and might serve as a barrier for researchers lacking monetary resources).

The threshold for Local Outlier Factor analysis is not mentioned.

The relative difference plots may benifit from reproduction to aid better legibility. Perhaps a dataflow diagram instead of the initial diagram to represent the overall working of the paper would help in making the paper more reader friendly.

**Reproducibility:**

3: Could reproduce the results with some difficulty. The settings of parameters are underspecified or subjectively determined; the training/evaluation data are not widely available.

**Reviewer Confidence:**

5: Positive that my evaluation is correct. I read the paper very carefully and I am very familiar with related work.

**Typos Grammar Style And Presentation Improvements:**

At times it does seem like the paper is not written in standard NLP research prose. However, given the empirical nature of the work, outside of a minor proofread and rewriting sentence to be more concise no additional work is needed for this work on its written work.
The images: the first one while relevant to outlier detection can be removed. The relative differences plots should be presented better as the numeric axis is hard to read and interpret.

---

> ### Author Rebuttal · Authors · 2023-08-29
>
> Thank you very much for your helpful feedback.
>
> **Novelty**
>
> Our paper introduces a method of identifying the harm a model causes toward different demographic groups through a new application of outlier detection. We operationalize outlier detection to identify data points that are unusual in three ways and quantifyies the amount of model error that is captured using this method. This method is unique because it translates the “other” examined extensively in social science literature into statistical terms.
>
> **Additional Model Analysis**
>
> In response to your concern that we cannot make broad claims about demographic differences due to only testing on one dataset and model, we will edit the language in the camera-ready version to clarify these claims. We have also reproduced our analysis using ELECTRA (TehranNLP-org/electra-base-hateXplain) and RoBERTa (facebook/roberta-hate-speech-dynabench-r4-target), which are available open source on HuggingFace, in addition to the Perspective API.
>
> In the RoBERTa and ELECTRA models, the weighted relative difference in mean squared error (WMSE) of text outliers compared to non-outliers was 0.056 and 0.036 points respectively, signifying a performance drop for outliers. Compared to many other demographic groups, both text and demographic outliers reveal greater disparities in model performance. Thus, we note that the performance drop on outliers is not necessarily revealed by a specific type of outlier, but rather that both demographic and text outliers reveal performance drops on different models. However, whereas the Perspective API uses the same definitions of toxicity as the Jigsaw dataset, the other models may use different definitions, so the results corresponding to the Perspective API are still more reliable. We will also add versions of Table 1 for the other two models.
>
> RoBERTa Percentiles:
>
> |                  | Demographic Outliers | **Text Outliers** | Disagreement Outliers |
> |------------------|-------------|----------|--------------|
> | _Marginalized_   | 33.3        | **77.8** | 44.4         |
> | _Binary_         | 51.9        | **92.6** | 66.7         |
> | _Intersectional_ | 21.1        | **95.0** | 83.9         |
>
> ELECTRA Percentiles:
>
> |                  | Demographic Outliers | **Text Outliers** | Disagreement Outliers |
> |------------------|-------------|----------|--------------|
> | _Marginalized_   | 44.4        | **88.9** | 33.3         |
> | _Binary_         | 74.1        | **81.5** | 59.3         |
> | _Intersectional_ | 83.2        | **85.7** | 79.6         |
>
> Due to the changes to 4.1, we will broaden 4.3 and 4.4 to discuss text outliers as well as demographic outliers. We conducted an identical analysis for text outliers and found that the results for these sections are similar for text and demographic outliers except for some differences in the demographic makeup of text vs. demographic outliers. Similar to the demographic outlier analysis, the models were significantly worse at predicting severe toxicity (68.4%) and identity attack (37.2%) for outliers compared to RoBERTa toxicity (35.2%), ELECTRA toxicity (22.9%), obscenity (15.1%), threat (14.3%), insult (8.34%), and Perspective toxicity (5.03%).
>
> Since mentions of race are overrepresented in text outliers while mentions of minor religions and sexual identities are overrepresented in demographic outliers, mitigation efforts must focus on groups with a high frequency among text outliers for the ELECTRA and RoBERTa models and demographic outliers for the Perspective models.
>
> **Commercial Model Limitations**
>
> We also acknowledge that using a commercial model may hinder reproducibility. We note that the Perspective API is free for all use cases. We use API version v1alpha1 for this analysis. According to the Perspective API’s changelog, the English-language models have not been updated since September 17, 2020. We will inquire with the Perspective team about current model version numbers, include that information in the camera-ready version, and note it as a limitation. The Jigsaw Unintended Bias in Toxicity Classification dataset is publicly available online through Kaggle and Huggingface; we will update the camera-ready version to clarify this and provide a link to the dataset.
>
> **Local Outlier Factor**
>
> In regards to the threshold for Local Outlier Factor, we use a contamination of 0.05, resulting in thresholds of -0.981 for demographic outliers, -0.989 for text outliers, and -0.982 for disagreement outliers.  The contamination value refers to the proportion of points selected as outliers. We specified a consistent contamination value instead of an explicit threshold when running LOF to have consistent proportions of outliers (5%) across the three outlier types. We will clarify this in the camera-ready version.
>
> **Figures**
>
> For the camera-ready version, we will address the comments about the diagrams by improving the relative difference plots and replacing our synthesis diagram.
>
> **Model and Data Selection**
>
> Regarding your question about how we chose the model and dataset for this work, we selected the dataset available publicly on Kaggle as Jigsaw Unintended Bias in Toxicity Classification dataset due to its detailed annotation of demographic labels using guidelines from the Online Hate Index Research Project at D-Lab, University of California, Berkeley. We chose to use the Perspective API because its models are calibrated with the scores in the dataset. This is not the case for the other models we added in our revisions.
>
> **Data Limitation Effects**
>
> We agree that performance is likely worse on outliers because, by definition, there is less data similar to outliers in the dataset. This can be viewed as an issue of limited data on some groups causing those data points to be outliers, which leads to worse performance on that data. We will further clarify this in the camera-ready version.

---

### Official Review · Reviewer_SNHy · 2023-08-06

**Soundness:** 4

**Excitement:**

4: Strong: This paper deepens the understanding of some phenomenon or lowers the barriers to an existing research direction.

**Paper Topic And Main Contributions:**

The authors introduce an important dimension in the direction of offensive/toxicity detection. It is also a challenging problem in the area as detecting the outliers is testing the capabilities of a model and can help with identifying the test/bounds of a model.

**Questions For The Authors:**

A. Did the authors look into the content empirically as to where their methods does do an improved prediction/detection of the outliers than a method which does not look into such case?
B. Numbers does show the improvement, but it would be beneficial for the readers to understand in a contextual sense with set of examples.

**Reasons To Accept:**

A. The authors have done a tremendous job of analyzing and stress testing their method. The paper is well written and easy to follow.
B. The list of experiments along with the other methods utilized for their evaluations does help with grounding their work.

**Reasons To Reject:**

More of questions than stronger reasons to reject.

**Reproducibility:**

5: Could easily reproduce the results.

**Reviewer Confidence:**

4: Quite sure. I tried to check the important points carefully. It's unlikely, though conceivable, that I missed something that should affect my ratings.

---

> ### Author Rebuttal · Authors · 2023-08-29
>
> Thank you very much for your feedback.
>
> We did not attempt to empirically identify outliers, but we will include an appendix with examples of text that are in and out of each outlier group and the associated scores to make our research more easily understandable to readers.

---

### Meta-Review · Area_Chair_pqDv · 2023-09-18

**Recommendation:** 4

**Metareview:**

This manuscript examines how toxicity detection methods impact marginalized communities using outlier detection. The reviewers are generally positive towards the manuscript, for instance, they note that 1) the manuscript is well written and organized, 2) the method presented is well tested, and 3) the experiments are useful towards grounding the point. Moreover, reviewers highlight that 4) the background research of the paper as particularly well done.

In terms of weaknesses, reviewers ask that a) the authors describe the dataset in more detail, b) the method is not novel, c) experimental details are missing, d) limited claims (due to single model tested).

In response to a) and c) the authors provide further details in their author response, which should be included in the manuscript. Responding to d) the authors show similar results for two additional models - which should similarly be included and discussed. In response to b) the authors highlight that while neither method nor data are new, they apply them together to identify a novel method for identifying harms to demographic groups, i.e. the use case and findings are novel.

The ethics reviewers ask that the authors address the issue of dual use of their method to identify outliers with malicious goals (i.e., to cause harm).

I would ask that the authors include a consideration on “norm” in their limitations, particularly what norm means and what the impact of assigning communities to be outside of the norm has as implications.

Some reference issues/suggestions.
On normalcy, Lepawsky, 2019 (https://discardstudies.com/2019/09/23/no-insides-on-the-outsides/) and Haraway (Situated Knowledges) of which speak to normalcy and positionality w/ Lepawsky specifically talking about content moderation and toxicity.
Thylstrup and Talat (Dirt and Toxicity) speak to norms in data (and subjectivity) and harms in the context of content moderation.
Kearns et al. 2018 (Fariness Gerrymandering) is useful reference for identifying subgroups and the issues that arise computationally.
Rather than Crawford 2017, the better reference there would be Butler’s Excitable Speech, which deals directly with the issue of harms arising from recognition (or to be identified) - particularly the introduction and chapter 1 would be useful for the authors.

---

### Decision · Program_Chairs · 2023-10-07

**Decision:**

Accept-Main

**Comment:**

This manuscript examines how toxicity detection methods impact marginalized communities using outlier detection. The reviewers are generally positive towards the manuscript, for instance, they note that 1) the manuscript is well written and organized, 2) the method presented is well tested, and 3) the experiments are useful towards grounding the point. Moreover, reviewers highlight that 4) the background research of the paper as particularly well done.

In terms of weaknesses, reviewers ask that a) the authors describe the dataset in more detail, b) the method is not novel, c) experimental details are missing, d) limited claims (due to single model tested).

In response to a) and c) the authors provide further details in their author response, which should be included in the manuscript. Responding to d) the authors show similar results for two additional models - which should similarly be included and discussed. In response to b) the authors highlight that while neither method nor data are new, they apply them together to identify a novel method for identifying harms to demographic groups, i.e. the use case and findings are novel.

The ethics reviewers ask that the authors address the issue of dual use of their method to identify outliers with malicious goals (i.e., to cause harm).

I would ask that the authors include a consideration on “norm” in their limitations, particularly what norm means and what the impact of assigning communities to be outside of the norm has as implications.

Some reference issues/suggestions.
On normalcy, Lepawsky, 2019 (https://discardstudies.com/2019/09/23/no-insides-on-the-outsides/) and Haraway (Situated Knowledges) of which speak to normalcy and positionality w/ Lepawsky specifically talking about content moderation and toxicity.
Thylstrup and Talat (Dirt and Toxicity) speak to norms in data (and subjectivity) and harms in the context of content moderation.
Kearns et al. 2018 (Fariness Gerrymandering) is useful reference for identifying subgroups and the issues that arise computationally.
Rather than Crawford 2017, the better reference there would be Butler’s Excitable Speech, which deals directly with the issue of harms arising from recognition (or to be identified) - particularly the introduction and chapter 1 would be useful for the authors.